# Modelling vegetation understory cover using LiDAR metrics

Lisa A. Venier[1]*, Tom Swystun[1], Marc J. Mazerolle[2], David P. Kreutzweiser[1], Kerrie L. Wainio-Keizer[1], Ken A. McIlwrick[1], Murray E. Woods[3], Xianli Wang[1]

1 Canadian Forest Service, Great Lakes Forestry Centre, Natural Resources Canada, Saul Ste Marie, ON, Canada, 2 Department of Wood and Forest Sciences, Center for forest research, Université Laval, Quebec, QC, Canada, 3 Ontario Ministry of Natural Resources and Forestry, North Bay, ON, Canada

* lisa.venier@canada.ca

**Data Availability Statement:** All relevant data are within the manuscript and its Supporting Information files.

## Abstract

Forest understory vegetation is an important characteristic of the forest. Predicting and mapping understory is a critical need for forest management and conservation planning, but it has proved difficult with available methods to date. LiDAR has the potential to generate remotely sensed forest understory structure data, but this potential has yet to be fully validated. Our objective was to examine the capacity of LiDAR point cloud data to predict forest understory cover. We modeled ground-based observations of understory structure in three vertical strata (0.5 m to < 1.5 m, 1.5 m to < 2.5 m, 2.5 m to < 3.5 m) as a function of a variety of LiDAR metrics using both mixed-effects and Random Forest models. We compared four understory LiDAR metrics designed to control for the spatial heterogeneity of sampling density. The four metrics were highly correlated and they all produced high values of variance explained in mixed-effects models. The top-ranked model used a voxel-based understory metric along with vertical stratum (Akaike weight = 1, explained variance = 87%, cross-validation error = 15.6%). We found evidence of occlusion of LiDAR pulses in the lowest stratum but no evidence that the occlusion influenced the predictability of understory structure. The Random Forest model results were consistent with those of the mixed-effects models, in that all four understory LiDAR metrics were identified as important, along with vertical stratum. The Random Forest model explained 74.4% of the variance, but had a lower cross-validation error of 12.9%. We conclude that the best approach to predict understory structure is using the mixed-effects model with the voxel-based understory LiDAR metric along with vertical stratum, because it yielded the highest explained variance with the fewest number of variables. However, results show that other understory LiDAR metrics (fractional cover, normalized cover and leaf area density) would still be effective in mixed-effects and Random Forest modelling approaches.

## Introduction

Understory vegetation is an important part of the forested ecosystem. It contributes greatly to nutrient cycling [1, 2], wildlife habitat [3–5], fire behaviour [6–8], microclimate [2] and carbon

**Funding:** This study was funded by the Canadian Wood Fibre Centre, Natural Resources Canada to LV. (https://www.nrcan.gc.ca/forests/resources/research-centres-and-forests/canadian-wood-fibre-centre/13457). The funders had no role in study design, data collection and analysis, decision to publish, or preparation of the manuscript.

**Competing interests:** The authors have declared that no competing interests exist.

accounting [9]. Understory vegetation communities are therefore often considered a good indicator of forest ecological integrity [10, 11]. However, spatial predictions of understory cover or density have been extremely difficult to generate using traditional variables such as topography, overstory and soils [12]. Active remote-sensing technology such as LiDAR (light detection and ranging) could be used to generate estimates to address this issue.

LiDAR provides an estimate of three-dimensional forest structure including estimates of canopy structure, understory vegetation and terrain. LiDAR is a survey method that measures the return time of a laser light pulse reflecting off solid objects such as the vegetation or the ground. These laser returns generate a three-dimensional representation of the forest. This capacity has conferred large advantages to forest managers, conservationists and researchers in their attempts to manage the forest efficiently and sustainably. LiDAR can generate reliable, robust estimates of many forest structure variables including canopy height and cover [13–15], as well as basal area and tree density [13, 16] and has similar potential for understory structure.

Our objective in this paper is to evaluate the potential of LiDAR to generate predictions of understory cover by comparing to field measures of understory. To achieve this objective, we examine alternative LiDAR metrics that control for spatial heterogeneity of sampling density, we compare regression and machine learning statistical approaches, and we examine the value of multiple variables in our models.

A key challenge of working with LiDAR data is that there is a large amount of spatial heterogeneity in the sampling density over space that occurs in the normal course of generating LiDAR point clouds. This spatial heterogeneity is due to variations in scan angle, flight height, movement of the aircraft during data collection, the degree of overlapping flight lines, and topography [17–20]. Thus, relative measures of vegetation density or cover, where the number of returns in a vertical stratum are scaled relative to some measure of sampling density, should provide better estimates of true understory vegetation cover. A variety of approaches have been used to relativize these measures, for example, dividing the number of returns in a vertical bin by the total number of returns in the column, or by the number of returns in the bin and below the bin [21]. We examine four different understory structure metrics based on different approaches to control for sampling density.

We explored two statistical approaches for modelling understory vegetation structure as a function of LiDAR data: machine learning and mixed effects regression models. Machine learning, specifically Random Forest [22], has been used to model forest inventory variables with a large suite of LiDAR derived predictors [23, 24]. Machine learning in this context strives to produce the best prediction of the forest inventory variables. However, machine learning does not produce an ecologically interpretable relationship per se, only estimates of variable importance. Machine learning makes no assumptions about the structure of the data, is ideal for predicting relationships that are non-linear, is insensitive to correlations among variables, and interactions are automatically modeled [25]. However, machine learning is prone to bias associated with incomplete ranges of conditions being sampled [25]. As an alternative, we explored linear mixed-effects regression models. These models make assumptions of homoscedasticity and normality of errors which must be checked but can produce more parsimonious and more interpretable models than machine learning in some instances. In Random Forest models, large suites of variables are usually included to achieve the best predictive capacity. In the regression models, it is more important to limit the number of variables included to avoid overfitting and strong correlations between explanatory variables.

Occlusion has been discussed in the literature as a possible issue limiting LiDAR effectiveness for prediction of understory structure [26, 27], but more recent studies have shown that the potential occlusion may not interfere with generating predictions. Latifi et al. [23] demonstrated that artificially reducing the density of the LiDAR point cloud did not have an

appreciable effect on variance explained in models predicting understory structure. In another study, prediction errors of understory vegetation cover were not related with canopy cover [28]. However, forest type in some instances can influence the predictive accuracy of models [29]. In both of our modelling approaches, we included additional variables beyond the understory LiDAR metrics that may influence the amount of occlusion of the laser pulse, namely, the amount of overstory, the forest type, and the vertical stratum. All three of these variables could reflect the amount of vegetation in the area above the vertical stratum of interest.

Our primary objective is to quantify the capacity of LiDAR to estimate understory structure. To achieve this, 1; we compare the effectiveness of four possible understory LiDAR metrics for predicting understory cover that control for sampling density, 2; we examine the influence of potentially important additional explanatory variables on the model which will inform us about the importance of occlusion, and 3; we compare the mixed effects vs Random Forest approach for generating predictions. Our aim is to generate robust and effective predictions of understory cover that could inform forest management and conservation.

## Methods

### Study area

This project was conducted in the Petawawa Research Forest. Permission to conduct the study at the Petawawa Research Forest was granted by Natural Resources Canada. The research forest covers 9,945 hectares in the Great Lakes-St. Lawrence forest region (45˚ 58' 46.74" N, 77˚ 30' 22.11" W), Ontario, Canada. The study area is on the Southern end of the Precambrian Shield, on bedrock of granites and gneisses. Forest composition features White Pine (*Pinus strobus* Linnaeus), Red Pine *(Pinus resinosa* Aiton), Red Oak *(Quercus rubra* Linnaeus), Yellow Birch (*Betula alleghaniensis* Britton), Sugar Maple *(Acer saccharum* Marshall), and Red Maple *(Acer rubrum* Linnaeus) as dominant species, often in uneven-aged forests. Presently, the Petawawa Research Forest is dominated by healthy but mature and overmature overstory (80–140 years) coupled primarily with low-quality regeneration and understories. For the purpose of the current study, we classified the forest into four types (TYPE) to explore the influence of forest type on the consistency of the relationship between understory vegetation structure measured in the field and LIDAR metrics. The four classes of forest type (TYPE) are Pine, Red Oak, Mixedwood without Pine, and Mixedwood with Pine. These four classes account for approximately 71% of the landbase of the research forest.

### Field data collection

Within the Petawawa Research Forest, plots were selected from a 25 m-resolution rasterized LiDAR database and Forest Resource Inventory data based on aerial photo interpretation. Potential plots were selected based on a stratification by forest type, overstory density, and understory density. Initial overstory was measured as the relative number of LiDAR laser pulse returns in overstory ($>$ 4 m), and understory density as the relative number of LiDAR laser pulse returns 4 m or lower. We divided the full range of overstory values into 10 equal bins, and the full range of understory values into 10 equal bins. For each combination of understory by overstory bin we selected five potential plots for each of four forest types, for a total of 2000 plots, 500 in each 10 by 10 matrix, with one matrix per each forest type. This is a rough stratification but helped to fill the statistical space to ensure optimal conditions for model construction. We sampled 437 plots out of the possible 2000, trying to select 1–5 plots from all cells in the matrix. We acknowledge that this stratification would not be effective if the relative number of LiDAR pulse returns was unrelated to actual understory vegetation cover. However, it

was the most intuitive method to ensure that all overstory and understory conditions in our study area were represented in the sample.

We collected vegetation data on 250 plots in 2015 and on an additional 187 plots in 2016. Plots were selected in the field from the list of preselected plots based on accessibility and conformity with classified forest type, understory, and overstory. At each plot centre, we used an SX Blue II GPS to generate a sub-meter accurate location through averaging a minimum number of 300 points (Geneq Inc., Montreal, Canada). Our field data collection attempted to generate a field-based point cloud to match the LiDAR based point cloud. We measured forest structure on ground-based plots in nine vertical strata (0–0.5 m, 0.5–1 m, 1–1.5 m, 1.5–2 m, 2–2.5 m, 2.5–3 m, 3–3.5 m, 3.5–4.0 m, > 4 m). From the centre point we created eight radial transects (12 m in length each) starting in a north direction and moving clockwise by 45 degrees for each additional transect. Along each transect, data were collected at each meter for a total of 97 sample locations in each plot, including the centre point (Fig 1). To sample the vegetation structure, observers recorded the presence or absence of vegetation within a radius of 15 cm for each of the nine vertical strata. Thus, there were 97 sampling points x 9 strata = 873 presence/absence points collected in each 12 m radius plot volume. The original vertical strata were later grouped into three strata (S1 = 0.5–1.5 m, S2 = 1.5–2.5 m, S3 = 2.5–3.5 m). We excluded points below 0.5 as they are difficult to distinguish from ground points. We excluded points above 3.5 m as they were difficult to estimate from the ground. The total number of vegetation presences in each stratum (0–194) were recorded in the FIELD variable for subsequent analysis. This field collection would represent a lower sampling density than the LiDAR data which are at 6 pulses per square meter with up to 8 returns per pulse which resulted in 2.44 returns per m$^3$ compared to the field data with 0.43 returns per m$^3$. These data are not strictly comparable since the field data represent presence and absence, whereas the LiDAR returns represent only presence but give a general impression of relative sampling density.

## LiDAR acquisition

Airborne LiDAR data were collected over the Petawawa Research Forest from August 17–20, 2012. The Riegl 680i sensor was carried aboard a Cessna 172 aircraft flown at an average altitude of 750 m. Technical acquisition specifications are provided in Table 1. The data were collected as a full-waveform and provided as a discrete point file (LAS 1.1) for use in this project. Flight overlap was approximately fifty percent.

## Data processing and LiDAR variables

We developed specific LiDAR understory cover metrics that are expected to capture the vegetation understory density directly. We identified four metrics for our analysis. Three of these metrics are used in the literature: fractional cover (FRAC, modified from Wing et al. [28]), leaf area density (LAD, [30]), and voxel cover (VOX1m, [31]). The fourth metric considered was normalized cover (NORM), because it is an easily interpretable and easily calculated alternative. Fractional cover is calculated by summing the number of LiDAR vegetation returns for each understory vertical stratum and dividing by the sum of understory and ground returns. Leaf area density is calculated as the negative log of the number of returns in a vertical stratum divided by all returns in and below the vertical bin and then divided by a constant. Normalized cover is calculated by dividing all vegetation returns in the understory stratum divided by all first returns. The voxel cover approach filters all returns by estimating presence/absence of returns in each standard voxel (in our case 1 m$^3$) in the vertical stratum. For example, a 2 m x 5 m x 5 m vegetation stratum that contains 50 1-m$^3$ voxels would have a voxel cover value between 0 and 50, equal to the number of voxels that contain vegetation. Sampling density is

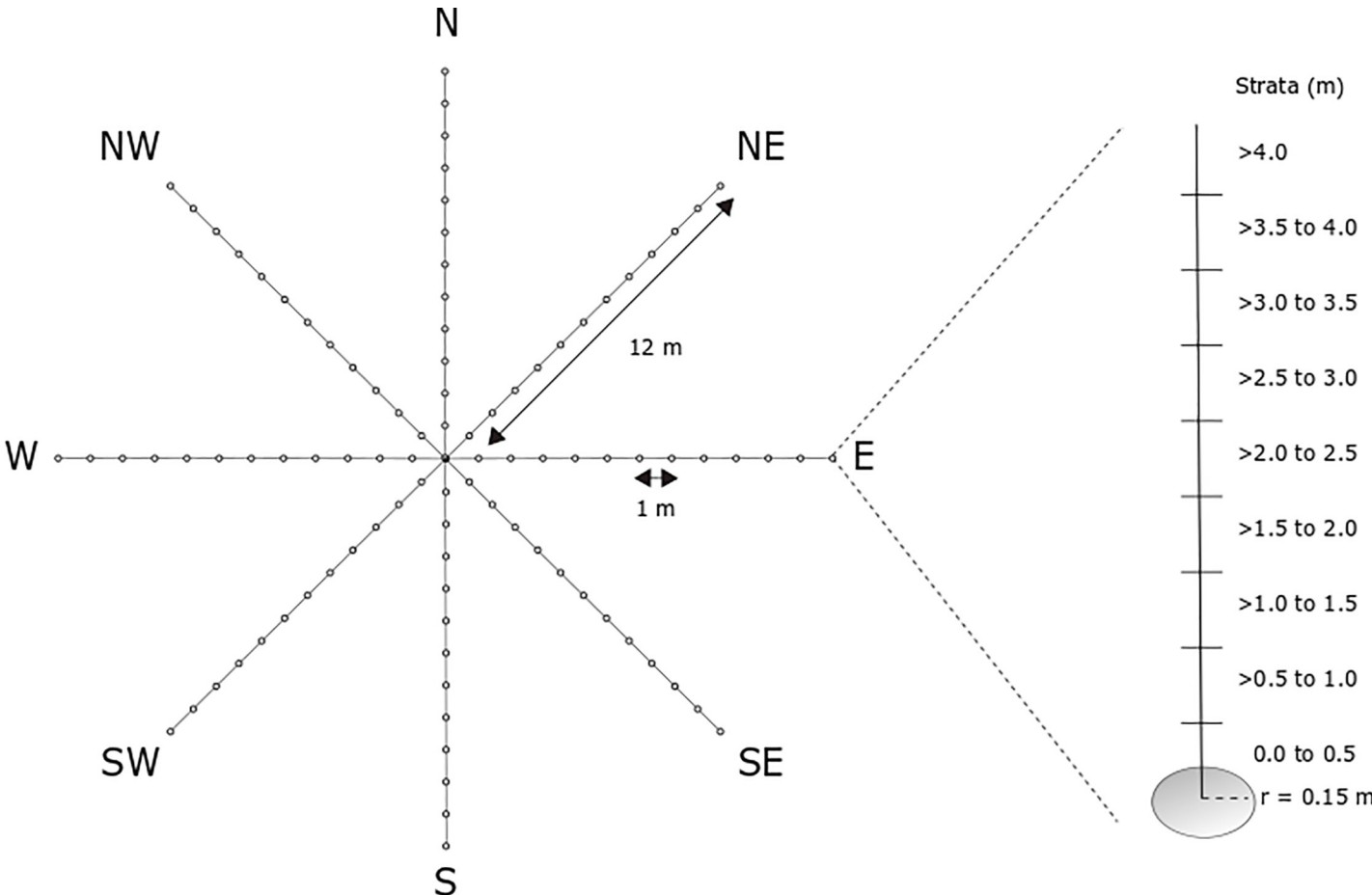

**Fig 1. Sampling design for field observations of vegetation structure (FIELD).** Measurements around each point on the transects and vertical strata were within a 15 cm-radius (r).

extremely heterogeneous due to different factors such as flight line overlap and the pitch and yaw of the plane. The LiDAR metrics provide four alternative ways to scale the number of returns in a vertical bin by sampling density. In addition to these four specific LiDAR

**Table 1. Airborne LiDAR acquisition specifications.**

| Parameter | Value |
|---|---|
| Pulse repetition rate | 150 Khz |
| Frequency | 76.67 Hz |
| Scan Angle | ± 20 Degrees |
| FOV | 40 Degrees |
| Line spacing: Cross track | 0.6 m |
| Line spacing: Along track | 0.6 m |
| Line spacing between flight lines | 250 m |
| Laser footprint min: | 0.38 m |
| Laser footprint max | 0.42 m |
| Average point density: All Returns | ~ 15 pts/m$^2$ |
| Average point density: Last Returns | ~ 6 pts/m$^2$ |

understory cover metrics, we calculated a suite of standard LiDAR point cloud metrics such as canopy cover and canopy height (S1 Table).

## Analysis

We used linear mixed effects models to determine the capacity of our four main LiDAR understory cover metrics to predict understory cover recorded in the field (FIELD) in each of the three vertical strata defined above (ST1, ST2, ST3), and to examine the influence of secondary explanatory variables [32]. These secondary explanatory variables consisted of forest TYPE (based on overstory composition), STRATUM (vertical 1 m strata, ST1-ST3), and OVERSTORY (S1 Table). The OVERSTORY variable was a measure of LiDAR vegetation cover in the vertical column above the stratum of interest calculated by classifying canopy cover (CC) into three classes (low, medium, high). We treated the plot as a random effect to account for multiple measurements in each plot. We formulated 16 candidate models consisting of LiDAR variables, with the constraint of maintaining variance inflation factors (VIF) < 10 to avoid issues of multicollinearity (Table 2). For each the four main LiDAR metric, we derived four models: 1) a null model consisting only of the LiDAR metric, 2) a model with the LiDAR metric, OVERSTORY and, their interaction, 3) a model with the LiDAR metric, TYPE, and their interaction, and 4) a model with the LiDAR metric, STRATUM, and their interaction. We ranked all mixed effects models based on Akaike's information criterion (AIC, [33, 34]) and calculated the $R^2$ values. We also computed the symmetric mean absolute percentage error (SMAPE), based on 10-fold cross-validation [35], for the top-ranked models, and calculated SMAPE values for each of the 3 vertical strata separately. Parameters of the mixed effects models were estimated by maximum likelihood in R with the nlme package [23, 32, 36].

We used Random Forest with the same FIELD response variable as in the mixed-effects models described above. Because Random Forests are non-parametric and do not yield a log-likelihood, we ran a stepwise procedure with 341 LiDAR derived variables (which includes overstory estimates) (S1 Table), plus secondary variables forest TYPE (from Forest Resource Inventory), and STRATUM. We used mean decrease in accuracy to rank variable importance [37]. At each iteration, we removed the 20% least influential variables and compared the explained variance. Models were built using the randomForest package in R [37]. We examined the importance of variables in the suite of random forest models. Similar to the mixed effects models above, we quantified model performance with the percent variance explained and SMAPE based on 10-fold cross-validation. Finally, we compared the prediction performance of the mixed effects and Random Forest approaches.

## Results

### Relationship among LiDAR metrics

The FIELD measure of understory cover was strongly correlated with all of the four main LiDAR metrics we investigated (Fig 2A–2D). However, the FRAC and VOX1m metrics were slightly more linearly correlated than the other metrics to the FIELD measure (Fig 2A–2D). Nonetheless, the four understory vegetation metrics were all highly correlated with one another (Table 3).

### Mixed-effects models

The model consisting of the voxel-based cover estimate (VOX1m) with STRATUM and their interaction was the most parsimonious among all sixteen models considered (Table 4). This model had all the support (Akaike weight = 1, Table 4, Fig 3A). This model also had the highest

**Table 2. Mixed effects model explaining understory cover recorded in the field (FIELD): TYPE = forest type based on overstory composition, STRATUM = vertical 1 m strata, ST1-ST3, and OVERSTORY = a measure of LiDAR vegetation cover in the vertical column above the stratum of interest calculated by classifying canopy cover (CC) into three classes (low, medium, high), see S1 Table.** The plot was treated as a random effect in each model.

| Model Name | Model fixed effects structure | Biological interpretation |
|---|---|---|
| FRAC null | FRAC | Relationship between FRAC and FIELD is constant |
| FRAC * STRATUM | FRAC + STRATUM + FRAC*STRATUM | Relationship between FRAC and FIELD differs among STRATUM |
| FRAC * OVERSTORY | FRAC + OVERSTORY + FRAC*OVERSTORY | Relationship between FRAC and FIELD differs among OVERSTORY |
| FRAC * TYPE | FRAC + TYPE + FRAC*TYPE | Relationship between FRAC and FIELD differs among TYPE |
| NORM null | NORM | Relationship between NORM and FIELD is constant |
| NORM * STRATUM | NORM + STRATUM + NORM*STRATUM | Relationship between NORM and FIELD differs among STRATUM |
| NORM * OVERSTORY | NORM + OVERSTORY + FRAC*OVERSTORY | Relationship between NORM and FIELD differs among OVERSTORY |
| NORM * TYPE | NORM + TYPE + FRAC*TYPE | Relationship between NORM and FIELD differs among TYPE |
| VOX1m null | VOX1m | Relationship between VOX1m and FIELD is constant |
| VOX1m * STRATUM | VOX1m +STRATUM + VOX1m*STRATUM | Relationship between VOX1m and FIELD differs among STRATUM |
| VOX1m * OVERSTORY | VOX1m + OVERSTORY + VOX1m*OVERSTORY | Relationship between VOX1m and FIELD differs among OVERSTORY |
| VOX1m * TYPE | VOX1m + TYPE + VOX1m*TYPE | Relationship between VOX1m and FIELD differs among TYPE |
| LAD (null) | LAD | Relationship between LAD and FIELD is constant |
| LAD * STRATUM | LAD + STRATUM + LAD*STRATUM | Relationship between LAD and FIELD differs among STRATUM |
| LAD * OVERSTORY | LAD + OVERSTORY + LAD*OVERSTORY | Relationship between LAD and FIELD differs among OVERSTORY |
| LAD * TYPE | LAD + TYPE + LAD*TYPE | Relationship between LAD and FIELD differs among TYPE |

conditional $R^2$ (along with the FRAC + STRATUM + interaction model, although all sixteen models had high $R^2$ values (0.71–0.87). For each of the four LiDAR metrics we considered, we observed the same pattern: the addition of STRATUM and the interaction to the null models resulted in consistently better model performance in terms of delta AIC and $R^2$. The addition of OVERSTORY or TYPE resulted in much less model improvement than the addition of STRATUM. The model with the most support did not include forest type or overstory, which is important since forest type was derived from forest inventory data and cannot be extracted from LiDAR point clouds.

The four LiDAR metrics had positive slopes in all of the mixed effects models (Fig 4, Table 5, for example). In our best model, the intercept of the lowest STRATUM was higher than in the upper strata (Fig 4). Although the model included the interaction between STRATUM and voxel cover, there was no evidence of different slopes of LiDAR among strata (Fig 4, Table 5). Symmetric mean absolute percentage (SMAPE) errors for the top-ranked mixed effects model was 0.156, but these values varied when investigating each stratum separately (Table 6). The SMAPE value was lowest for the lowest strata (0.107) and greatest for the highest strata (0.190) suggesting no evidence of occlusion. There were 437 observations for each stratum.

## Random forest models

We examined the percent variance explained and the number of variables included to choose a final Random Forest model. The base model with all 341 LiDAR-derived variables, forest TYPE, and STRATUM explained 74.8% of the variance, but the final model with only 59 predictors had a very similar variance explained (74.4%) (Fig 3B, Table 7, S2 Table). The 10-fold cross-validation on this reduced model showed an overall mean error rate of 0.129 (Table 6).

Some variables appeared more often than others among the 18 Random Forest models considered. These variables consisted of STRATUM, GAP (the inverse of LAD), and LAD. In addition, most or all of the LiDAR understory vegetation cover metrics (VOX1m, FRAC,

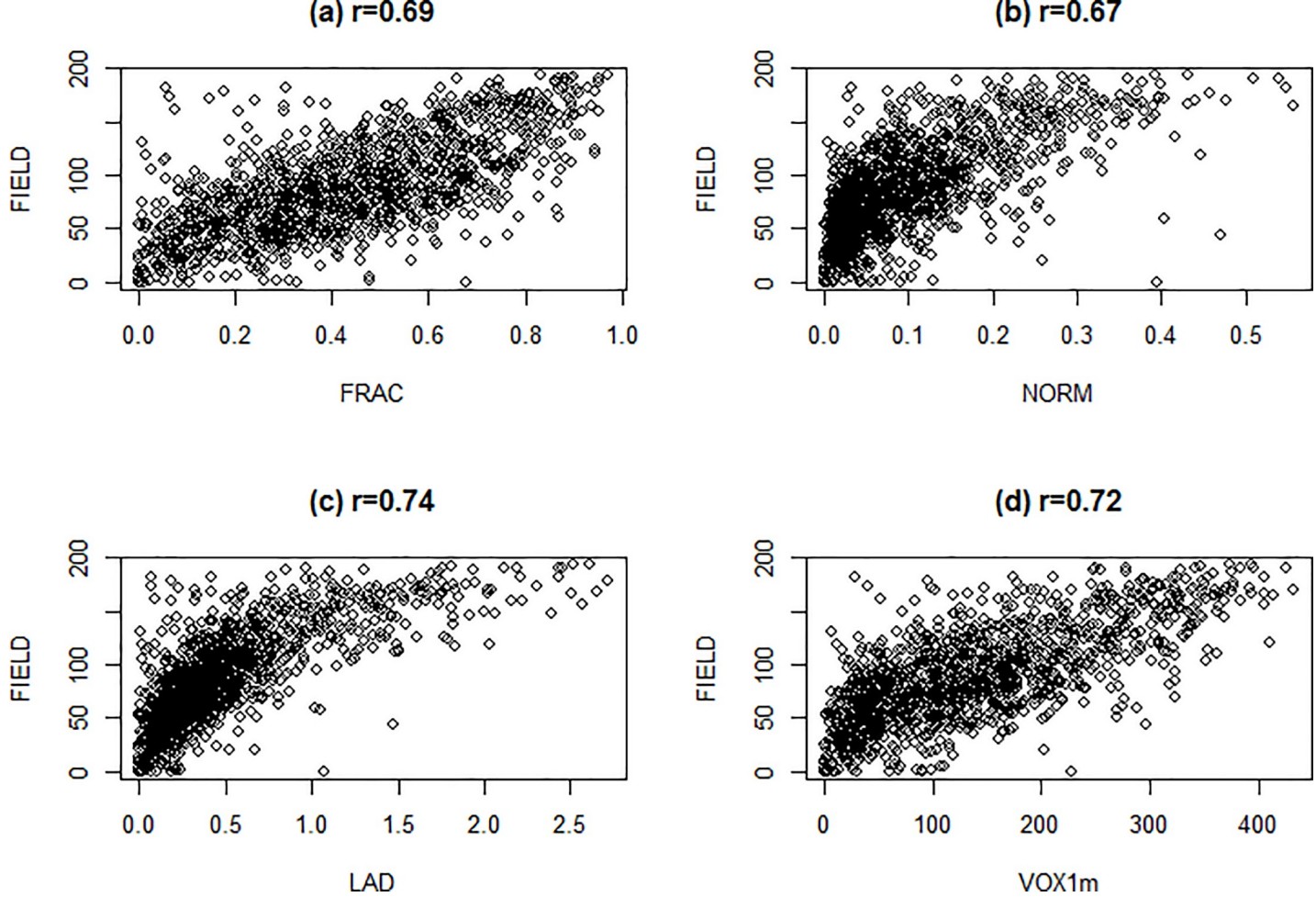

**Fig 2. Scatterplot of FIELD (measured density) against the LiDAR metrics**, a) fractional cover (FRAC), b) normalized cover (NORM), c) leaf area density (LAD), and d) voxel cover (VOX1m), including Pearson product-moment correlation coefficients.

NORM) were represented in the top 10 variables of most of the 18 potential models (S3 Table). Crown closure (CC), an estimate of overstory, was also often among the top 10 most important variables within the models considered. Forest TYPE never occurred among the top 10 variables (S3 Table).

**Table 3. Pearson product-moment correlations between pairs of understory cover LiDAR metrics included in analysis (n = 1310).**

| Correlation | r | Lower 95% CL | Upper 95% CL |
|---|---|---|---|
| FRAC vs NORM | 0.77 | 0.751 | 0.794 |
| FRAC vs VOX1m | 0.84 | 0.819 | 0.852 |
| FRAC vs LAD | 0.77 | 0.744 | 0.789 |
| NORM vs LAD | 0.81 | 0.79 | 0.827 |
| NORM vs VOX1m | 0.92 | 0.911 | 0.927 |
| VOX1m vs LAD | 0.79 | 0.767 | 0.808 |

**Table 4. $R^2$ and AIC values for sixteen candidate linear mixed-effects models.** Note that marginal $R^2$ denotes the percent variance explained by the fixed effects, whereas the conditional $R^2$ includes both fixed effects and random effects. Delta AIC is the difference between each model relative to the most parsimonious model and Akaike weight indicates the percent support of a given model.

| Model | Marginal $R^2$ | Conditional $R^2$ | AIC | Delta AIC | Akaike weight |
|---|---|---|---|---|---|
| VOX1m * STRATUM | 0.62 | 0.87 | 11868.87 | 0 | 1 |
| FRAC * STRATUM | 0.65 | 0.87 | 11901.00 | 32.13 | 0 |
| LAD * STRATUM | 0.56 | 0.82 | 11998.29 | 129.42 | 0 |
| NORM * STRATUM | 0.52 | 0.83 | 12099.16 | 230.29 | 0 |
| VOX1m * OVERSTORY | 0.60 | 0.82 | 12348.32 | 479.45 | 0 |
| LAD * OVERSTORY | 0.51 | 0.73 | 12384.88 | 516.01 | 0 |
| VOX1m * TYPE | 0.60 | 0.82 | 12384.88 | 516.01 | 0 |
| VOX1m null | 0.60 | 0.82 | 12385.78 | 516.91 | 0 |
| LAD * TYPE | 0.51 | 0.72 | 12396.42 | 527.55 | 0 |
| LAD null | 0.50 | 0.71 | 12407.11 | 538.24 | 0 |
| NORM * OVERSTORY | 0.53 | 0.75 | 12450.66 | 581.79 | 0 |
| NORM * TYPE | 0.51 | 0.75 | 12563.97 | 695.1 | 0 |
| NORM null | 0.49 | 0.75 | 12568.4 | 699.53 | 0 |
| FRAC * OVERSTORY | 0.58 | 0.77 | 12585.04 | 716.17 | 0 |
| FRAC * TYPE | 0.57 | 0.75 | 12613.19 | 744.32 | 0 |
| FRAC null | 0.56 | 0.75 | 12617.05 | 748.18 | 0 |

## Discussion

In this study, our primary objective was to quantify the capacity of LiDAR to estimate understory structure so that it can be predicted across a landscape. To address this objective, first we compared the effectiveness of four possible understory LiDAR metrics (fractional cover, leaf area density, voxel cover, and normalized cover) for predicting understory cover. Each of these metrics used some measure of the number or presence of LiDAR returns in an understory vertical stratum and standardized these measures with an estimate of sampling density. All four LiDAR metrics were effective at predicting the amount of structure in an understory stratum, probably because they are all highly correlated direct measures of the density of understory vegetation. The best metric based on mixed effects modelling, however, was the voxel-based cover estimate (VOX1m) with the addition of STRATUM with a conditional $R^2$ of 0.87. The voxel-based approach is relatively easy to calculate and provides a direct measure of the amount of understory structure.

We anticipated that other variables could influence the predictions of understory. We identified three potentially important variables that might influence occlusion of understory structure: overstory, forest type and stratum. Increased overstory can reduce the ability of LiDAR to predict understory structure due to occlusion [26, 27]. For LiDAR to detect the understory structure, LiDAR pulses must reach and be reflected by understory vegetation. A greater vegetation interception above the area of interest will result in fewer pulses returning from the understory. Both forest type and stratum will also influence the amount of vegetation in the area above the area of interest and therefore potentially alter the relationship of field measured and LiDAR measured understory.

Correlations between the three secondary explanatory variables (STRATUM, forest TYPE, and OVERSTORY) made it impossible to include all variables in a single model. Our best supported model included STRATUM, where we found that the lowest stratum (ST1, 0.5–1.5 m) had the highest intercept. This is consistent with occlusion in that we have more vegetation in ST1 than ST2 (1.5–2.5 m) and ST3 (2.5–3.5 m) for a given value of VOX1m. This is consistent

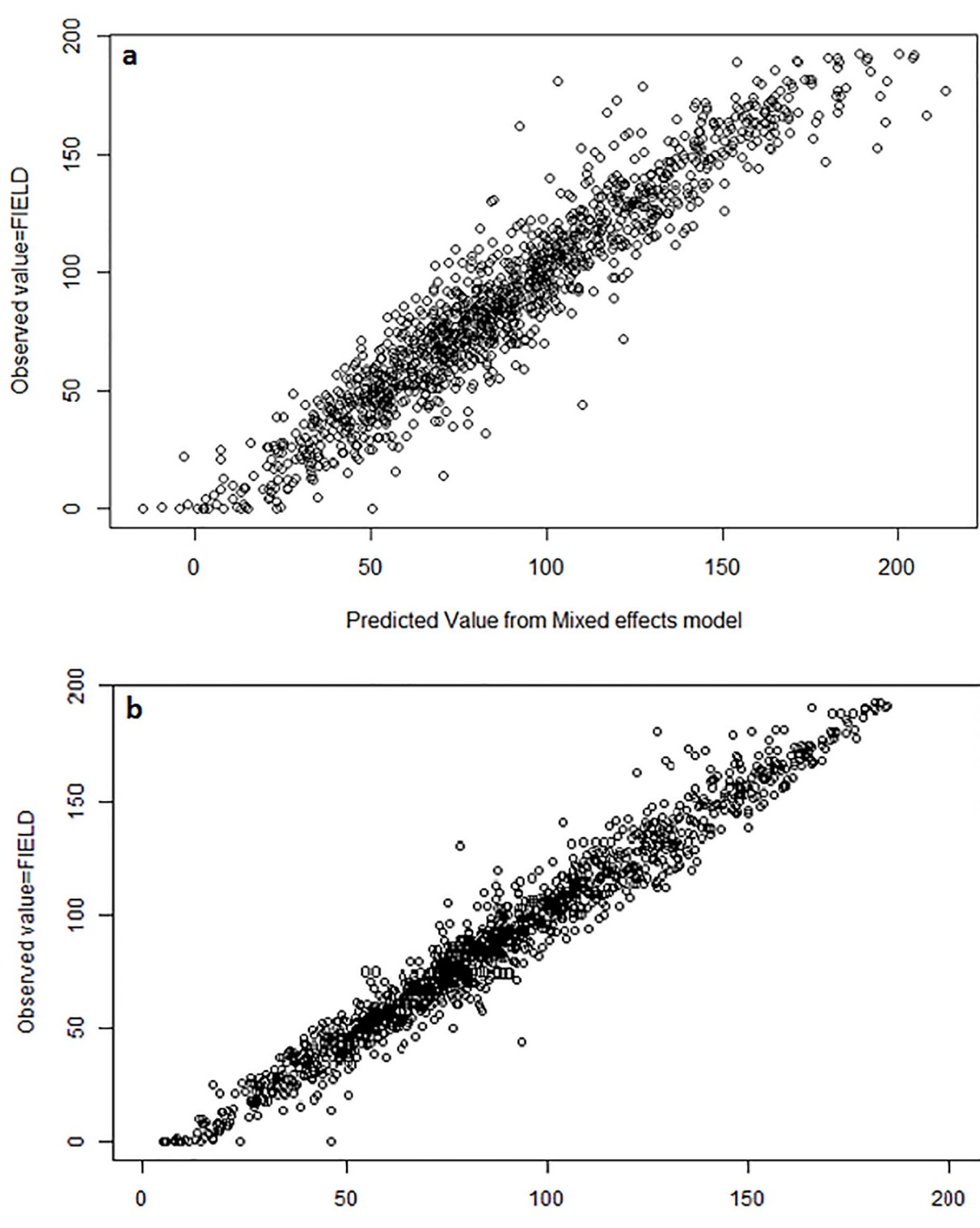

**Fig 3. Predicted versus observed scatterplot.** (a) Predictions of FIELD generated from mixed-effects model consisting of VOX1m + STRATUM + interaction, (b) Predictions of FIELD generated from Random Forest model with 59 explanatory variables.

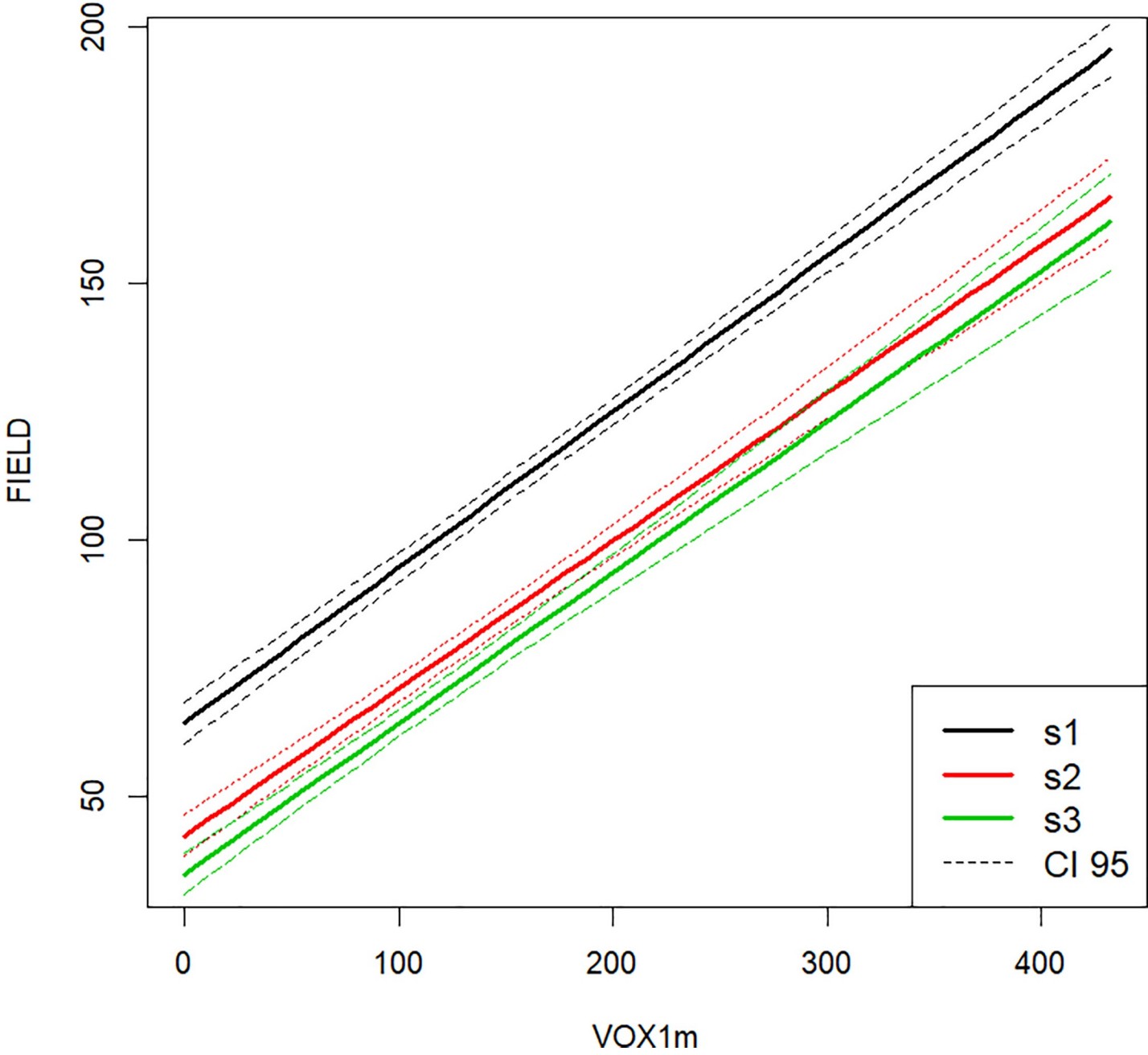

**Fig 4. Predictions of FIELD for each of three strata based on the mixed-effects model consisting of VOX1m + STRATUM + interaction.** Dashed lines around solid lines denote 95% confidence intervals around predictions.

with the idea that fewer laser pulses are reaching the lower stratum. The relationship between the field observed structure and VOX1m did not vary with STRATUM. Surprisingly, we found that the error in the predicted relationship was greatest in the highest STRATUM and lowest in the lowest STRATUM suggesting that there was no reduction in predictability associated with potential occlusion. These differences in prediction error suggest that the model can better predict new observations in the low stratum than the high stratum. A potential explanation for this result would be that the understory vegetation in the lower stratum is easier to estimate

**Table 5. Estimates of the best supported mixed-effects model consisting of VOX1m + STRATUM + interaction and a random effect of plot.**

|  | Estimate | Lower 95% CL | Upper 95% CL |
|---|---|---|---|
| intercept | 64.35 | 60.25 | 68.46 |
| LIDAR | 0.03 | 0.29 | 0.32 |
| STRATUM.ST2 | -21.94 | -25.96 | -17.98 |
| STRATUM.ST3 | -29.38 | -33.48 | -25.28 |
| LIDAR*STRATUM.ST2 | -0.016 | -0.039 | 0.008 |
| LIDAR*STRATUM.ST3 | -0.010 | -0.037 | 0.017 |

on the ground and therefore there is less noise in the relationship between the field and the LiDAR measures in the lower stratum. Either way, we conclude that our LiDAR sampling intensity was sufficient in our forest system to capture the understory structure regardless of the density of vegetation above the area of interest and the related potential for occlusion.

There is some discrepancy in the literature on the effect of occlusion. Latifi et al. [29] found that thinning LiDAR data by artificially reducing the sampling density did not impact the effectiveness of models to predict understory. Their original data had a high point density of 30–40 points per $m^2$ and a maximum of 11 returns. Data were thinned to two different levels but Latifi et al. [29] do not report on the final point density after thinning. Our data are at roughly 11.69 vegetation returns per $m^2$, with about 0.55 vegetation returns per $m^3$ in the 0.5–4 m understory stratum. Obviously, the effectiveness of LiDAR to capture understory structure will eventually be undermined by a sufficient reduction in sampling density, but this limit does not seem to have been reached in the Petawawa research forest. Gonzalez-Ferreiro et al. [38] showed that reducing pulse density from 8 pulses per $m^2$ to 0.5 pulses per $m^2$, did not decrease model precision in estimating stand variables. Wing et al. [28] found no trends between understory vegetation cover prediction error and canopy cover, lending support to the idea that under some natural overstory conditions and common LiDAR sampling densities, occlusion is not an issue for predicting understory with LiDAR. In contrast, Ruiz et al. [19] reported an effect of LiDAR sampling density on model $R^2$ values but only at levels below around 5 points/$m^2$. It is unclear how this number translates into pulses reaching the understory. The lack of influence of forest type on understory cover predictions enables predicting understory from LiDAR alone without relying on traditional forest resource inventory data.

The comparisons of mixed effects and Random Forest models revealed some obvious alignment. All four of the LiDAR metrics considered (fractional cover, leaf area density, normalized cover, and voxel cover) produced models with high $R^2$ values. All four of these variables also had very high variable importance in the Random Forest models. Voxel cover (VOX1m) was the most important variable in the selected Random Forest Model. The stratum variable appeared often in the top Random Forest models and was also important in the top-ranked mixed-effects model (VOX1m * STRATUM). The Random Forest model had a high variance

**Table 6. Ten-fold cross-validation results from top linear mixed-effects model and the selected Random Forest model, based on symmetric mean absolute percentage error (SMAPE).** Note that average values of SMAPE are given for predictions of all STRATUM levels, but also for predictions specific to STRATUM levels.

| Model |  | SMAPE mean | SMAPE sd (n = 10) |
|---|---|---|---|
| VOX1m * STRATUM | predictions of all STRATUM levels | 0.156 | 0.014 |
|  | predictions of STRATUM 1 | 0.107 | 0.016 |
|  | predictions of STRATUM 2 | 0.170 | 0.024 |
|  | predictions of STRATUM 3 | 0.190 | 0.020 |
| Random forest (59 predictors) |  | 0.129 | 0.015 |

**Table 7. Random forest models: Mean squared residuals and percent variance explained.**

| Number of Predictors in model | Mean Squared Residuals | Percent variance Explained |
|---|---|---|
| 341 (Base model) | 484 | 74.8 |
| 276 | 485 | 74.7 |
| 223 | 485 | 74.8 |
| 180 | 484 | 74.7 |
| 145 | 476 | 75.2 |
| 116 | 486 | 74.7 |
| 93 | 481 | 75.0 |
| 74 | 492 | 74.3 |
| 59 | 490 | 74.4 |
| 47 | 513 | 73.3 |
| 37 | 508 | 73.5 |
| 29 | 531 | 72.4 |
| 22 | 553 | 71.2 |
| 17 | 528 | 72.5 |
| 13 | 558 | 70.9 |
| 10 | 580 | 69.8 |
| 7 | 569 | 70.4 |
| 5 | 632 | 67.1 |

explained (75%), but not as high as the best mixed effects model that included the voxel-based measure of cover (87%). Our selected Random Forest model had 59 explanatory variables, whereas the best mixed effects model had two explanatory variables and their interaction, as well as a random effect of plot. Other variables with high importance in the Random Forest models included other direct measures of understory structure, and canopy closure (S2 Table), which is expected to influence the amount of vegetation in the understory through light availability. The prediction error was slightly lower for the random forest model than for the mixed effects model (12.9% vs 15.6%), albeit at the cost of including 59 explanatory variables compared to 8 parameters estimated in the mixed effects model. Based on our results, generating landscape-wide predictions using the mixed-effects model should be simpler and more efficient than with the Random Forest model. For these reasons (12% higher explained variance, fewer explanatory variables, and similar prediction error), we recommend the mixed effects model for predicting understory vegetation structure with LiDAR, but we acknowledge that the Random Forest model also generates robust predictions.

Direct evaluations of LiDAR metrics to capture understory cover are relatively rare. Studies have shown good agreement between field and LiDAR measures of forest stand biomass [39], but biomass is likely driven primarily by tree biomass rather than understory. Asner et al. [40] explored structural transformation of rain forests due to invasive plants and used LiDAR to estimate structural changes in the understory. However, Asner et al. [40] did not report quantitative comparisons of field and LiDAR measures. Martinuzzi et al. [41] produced classification accuracies of 83% in predicting the presence of shrubs, but not their abundance. Wing et al. [28] compared understory vegetation cover and airborne LiDAR estimates with the addition of a filter for intensity values in an interior ponderosa pine forest. Their models had $R^2$ values from 0.7 to 0.8 and accuracies of ± 22%. Our models achieved slightly higher $R^2$ with slightly lower error rates without the use of the intensity filter, suggesting that the latter filter may not always be necessary to generate good estimates. As well, the intensity filter is affected by a number of factors such as elevation and the nature of the object intercepted that are difficult to

normalize, so we prefer models that do not require intensity filters. Latifi et al. [29] also made a direct comparison of ground-based vs LiDAR estimates of understory cover in temperate mixed stands, and found strong relationships in the top canopy and the herbal layer with lower predictive power in the intermediate stand layers. Their shrub layer regression model had a relatively low $R^2$ value of 37%. In a later study, Latifi et al. [23] showed an $R^2$ of 80% for the shrub layer based on thinned LiDAR point clouds and new analytical methods. Campbell et al. [21] also compared field and LiDAR measures of understory directly in mixedwood forests and generated an $R^2$ of 0.44 based on a relative point density similar to metrics that we used here.

It is unclear why there is so much variation in the ability of LiDAR to predict understory structure but it suggests that we should be somewhat cautious in assuming that individual LiDAR metrics are always capturing the understory structure. It is important to note that some of the error in prediction in our models is likely the result of the lag between the LiDAR acquisition (2012) and the field data acquisition (2016–2017). This lag is likely to result in the most error in the youngest stands where changes in herb and shrub growth are likely to be greatest but I in the analysis, most stands are mature forest. Likely with less lag between LiDAR and ground-based measures we would have seen even better predictions. In addition, the error associated with GPS locations can introduce error into the relationship between ground-based and LiDAR estimates, although GPS technology is constantly improving. Our GPS (SXblue), reports sub meter accuracy under ideal conditions, but discrepancy in geoposition probably accounts for some of the error in prediction.

Despite the limited work directly evaluating LiDAR measures of understory vegetation structure, many studies have explored the use of LiDAR to capture wildlife habitat structure some of which is related to understory [42–46] One of the most commonly reported relationships is between vegetation structural diversity or understory density and wildlife diversity [5, 47–49]. In addition, vegetation understory structure explained bird species composition in a number of studies [5, 50, 51]. Melin et al. [52] found that a LiDAR metric similar to fractional cover to estimate shrub density below 5 m was a good predictor of grouse brood occurrence in Finland, consistent with expectations based on known habitat preferences of the species. However, they did not test the assumption that the LiDAR metric effectively estimates vegetation density below 5 m. All of these studies do however, provide indirect evidence for the effectiveness of LiDAR estimates to predict understory cover or density.

## Conclusions

Based on the highest variance explained, the fewer number of explanatory variables, and ease of interpretation and application, we recommend using the mixed-effects model consisting of voxel-based cover estimate, stratum, and their interaction to generate spatial estimates of understory cover. Nonetheless, all four LiDAR metrics that we considered and both analytical approaches (mixed effects models, Random Forests) produced predictions suitable for many ecological and forest planning applications. This information could improve spatially-explicit mapping of wildlife habitat, fire behaviour, or forest ecosystem dynamics. Measuring understory cover *in situ* is not difficult, but many applications require maps or spatial estimates of attributes for forest management and conservation applications over large areas. LiDAR remote sensing is the most efficient approach to generating these spatial estimates of forest attributes. Our results fully support the indirect evidence provided from wildlife studies that LiDAR can predict understory vegetation structure even in the presence of a mature tree canopy. With error percentages of around 15%, these spatial predictions will introduce some uncertainty into predictions, which should be factored into decision-making. With increasing

sampling density associated with better LiDAR technology, we anticipate that understory cover models will become more reliable and generalizable across regions. In particular, because the models are not dependent on any ecological relationships *per se*, because they use direct measures of vegetation cover, we believe that under similar sampling densities the models should be generalizable. Additional testing of this approach in different forested ecosystems would provide more confidence in the transferability of the models.

## Supporting information

**S1 Table. Definitions of all variables included in at least one of the mixed-effects or Random Forest models.**
(DOCX)

**S2 Table. Rank importance of explanatory variables in the selected Random Forest model with 59 variables.**
(DOCX)

**S3 Table. Frequency of explanatory variables among the 18 Random Forest models run with 341 to 7 variables.**
(DOCX)

**S1 Data. Data used in analyses for manuscript.** Variable definitions are found in S1.
(ZIP)

## Acknowledgments

N. Coops and P. Tompalski provided guidance and training on LiDAR data handling. P. Arbour and staff at the Petawawa Research Forest provided logistic support.

## Author Contributions

**Conceptualization:** Lisa A. Venier, David P. Kreutzweiser, Murray E. Woods.

**Data curation:** Tom Swystun, Kerrie L. Wainio-Keizer, Ken A. McIlwrick.

**Formal analysis:** Lisa A. Venier, Tom Swystun, Marc J. Mazerolle.

**Funding acquisition:** Lisa A. Venier.

**Investigation:** Kerrie L. Wainio-Keizer, Ken A. McIlwrick.

**Methodology:** Tom Swystun, David P. Kreutzweiser, Kerrie L. Wainio-Keizer, Ken A. McIlwrick, Murray E. Woods.

**Project administration:** Lisa A. Venier.

**Resources:** Xianli Wang.

**Visualization:** Tom Swystun, Murray E. Woods.

**Writing – original draft:** Lisa A. Venier.

**Writing – review & editing:** Lisa A. Venier, Tom Swystun, Marc J. Mazerolle, David P. Kreutzweiser, Murray E. Woods, Xianli Wang.

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
