## [Decision Letter · Decision Letter 0]

30 Aug 2019

PONE-D-19-19180

Modelling vegetation understory cover using LiDAR metrics

PLOS ONE

Dear Venier,

Thank you for submitting your manuscript to PLOS ONE. After careful consideration, we feel that it has merit but does not fully meet PLOS ONE’s publication criteria as it currently stands. Therefore, we invite you to submit a revised version of the manuscript that addresses the points raised during the review process.

ACADEMIC EDITOR: I agree with both reviewers that the study was well done, of interest to the broader scientific community, and requires on minor revision before acceptance for publication.

We would appreciate receiving your revised manuscript by Oct 14 2019 11:59PM. To enhance the reproducibility of your results, we recommend that if applicable you deposit your laboratory protocols in protocols.io, where a protocol can be assigned its own identifier (DOI) such that it can be cited independently in the future. For instructions see: http://journals.plos.org/plosone/s/submission-guidelines#loc-laboratory-protocols

We look forward to receiving your revised manuscript.

Kind regards,

John Toland Van Stan II, Ph.D.

Academic Editor

PLOS ONE

Journal Requirements:

Reviewers' comments:

Reviewer's Responses to Questions

**Comments to the Author**

1. Is the manuscript technically sound, and do the data support the conclusions?

Reviewer #1: Yes

Reviewer #2: Yes

2. Has the statistical analysis been performed appropriately and rigorously? 

Reviewer #1: Yes

Reviewer #2: Yes

3. Have the authors made all data underlying the findings in their manuscript fully available?

Reviewer #1: Yes

Reviewer #2: Yes

4. Is the manuscript presented in an intelligible fashion and written in standard English?

Reviewer #1: Yes

Reviewer #2: Yes

5. Review Comments to the Author

Reviewer #1: General Comments

I enjoyed reading the manuscript and thought it well done. The comparison of models is interesting and the results are interesting and unique. I think the specific comparison of model input variables has the potential to reinforce the findings and provide some deeper ecological insight (see below). For example, which variables show up in both top ranking models? Why do you think that is in the system you examined?

I think placing figures 3 and 5 together (one below the other) would be useful to really compare the differences…

Figure 4 needs larger text and more contrast, perhaps with different line patterns. (line type in R). It is hard to read.

Hypotheses need to be included in the final paragraph of your introduction. You make reference to them in the discussion.

I would like to see some commentary on why you think the variables that floated to the top in both models were what they were. What do you think their ecological significance is? Or perhaps, why that understory structure shows up that way in the data. Do you think these models or indices would be relevant in other forests or ecosystems?

Specific Comments

24 “… among other things” sounds very weak. I suggest changing this sentence to get a reader’s attention. Suggestion: “Forest understory vegetation is an important characteristic of the forest, but hard to measure with current remote sensing tools.”, or something along those lines.

40 If the random forest model had lower error, why did you choose the mixed effects model? Hope to return to this later.

52 remove the word “potentially”, it is redundant.

53 I would qualify this statement… Lidar itself is just data, the estimates come when models are created, which is what you are doing. Therefore, Lidar “can provide estimates” or something like that would be more apt. As you know, understory vegetation is typically obscured by the dominant canopy, which is why your models would be useful.

56-58 The most important measurement here is the time of flight of the reflected pulse, not just the pulse itself, which would give you the intensity of the return, so that piece of information in the third sentence here should come earlier. I think these sentences could be combined into a more concise description.

68-72 I don’t think you need to state these things here, as you repeat them in the remaining part of the introduction. If anything, these things should come at the very end of the introduction when you are tying everything together that you have presented thus far.

77 Heterogeneity is also caused by topography and the vegetation structure itself. And you should definitely have some citations here. Here’s one off the top of my head:

Goodwin, N. R., Coops, N. C., & Culvenor, D. S. (2006). Assessment of forest structure with airborne LiDAR and the effects of platform altitude. Remote Sensing of Environment, 103(2), 140-152.

82 I would include other citations here… there have been numerous efforts to normalize lidar point density. For example:

Ruiz, L., Hermosilla, T., Mauro, F., & Godino, M. (2014). Analysis of the influence of plot size and LiDAR density on forest structure attribute estimates. Forests, 5(5), 936-951.

And take a look at this one:

Jakubowski, M. K., Guo, Q., & Kelly, M. (2013). Tradeoffs between lidar pulse density and forest measurement accuracy. Remote Sensing of Environment, 130, 245-253.

92 You need a citation here concerning machine learning.

93 Also a citation here.

96 Citation here

99 And citation here. To make declarative statements about model parameters and interpretability like these, you should be referencing something.

118 What hypotheses do you have concerning variable importance, model fit, etc. that you could reference in the discussion?

144 I’d like to see something here concerning the justification for stratifying your plot locations based on the data that you are trying to predict with… it seems somewhat circular. I think it is ok, as that is really the only way you could come up with a stratification based on understory across a landscape, but you should still address this issue.

154 What was the mean/variation of the horizontal precision of your GPSed plot coordinates? Sub meter could mean 0.01m or 0.99m

163 The diagram is very helpful in understanding the plot design.

180 Could you justify the temporal discrepancy here? I am wondering how much the understory might have changed in the intervening 3 and 4 years between lidar acquisition and data collection.

189 I like this section. Easy to follow.

256 “appeared” is not a precise word. Perhaps something like “were slightly more linearly correlated”. The difference in correlation values is very small and I would assume not significant in a statistical sense.

272 “For each of the four…”

275 This is an interesting finding!

296 This should be referencing back to your initial hypotheses, but you didn’t present any in your final introductory paragraph.

299 The labels on this figure should be larger.

321 based on the lower error rate and looking at the scatter in Figure 5, I would say that the random forest model did a much better job in predicting understory strata.

331 I would like to see the ranking of the most important variables in your final random forest model.

345 This is from your mixed effects model. I personally think the random forest model did a better job of prediction, although as you state in the introduction, not as easy to interpret as the linear model. But when using such rich data and derivatives as possible with lidar, you might as well have a model that includes all relevant information, such as a random forest model (in my opinion, you might disagree). Perhaps here though, you could compare which variables were included in your top ranking linear models to the most important ranked variables in your final random forest model. Is there overlap? I think this would not only serve to justify the models and variables, but to compare what the models themselves say about the relevant predictor variables.

348 Your hypothesis should come before the discussion and reference them here.

389 You don’t mention random forest models up until this point. All of your discussion thus far comes off as there only being one model type explored… I would mention random forest earlier and like you did for mixed effects, discuss the final model and the most important variables within that model.

396 Variance explained doesn’t matter as much as the error… the random forest model had a lower rate of error and that should be stated here to counterbalance the statement about variance.

428-431 Yes! This is what I was thinking when I read the methods. I think you need to expand on this. Would your results been similar if they were collected during the same year? I think this is a really important point and shouldn’t just come at the end of a paragraph.

443 You talk a lot about the data and the models… I would like to see some hypothesis building about why certain metrics were better and how they are directly related to the structure.

443 I would also like to know, as a scientist, how you think these models, indices, metrics, etc. might work in other forests? Do you think they are particular to this ecosystem?

Reviewer #2: PLOS ONE – Manuscript ID: PONE-D-19-19180

Title: Modelling vegetation understory cover using LiDAR metrics

Reviewer comments:

Overall comments:

This is a well-written paper with logical flow that is easy to follow, has sufficient detail for replication by others, and use of terminology and acronyms is appropriate for audience. It is obvious a lot of hard work went into this project, and this paper does a great job of explaining it. I am especially impressed by the thorough lit review and excellent figures and tables, which some authors don’t put a lot of effort into. I have a few very minor comments, below.

Abstract

Line 27: has yet to be fully validated.

Introduction

Line 86 (and as it occurs thereafter): some papers cite Random Forests with capital letters. I think it is fine either way.

Excellent lit review. The authors did an especially good job exploring various options for analysis.

Methods

Line 126: extra space between composition features

Thorough Methods section. I like Figure 1.

Analysis

Line 214: insert space between variables (26)

Line 229: insert space between package (18…

Thorough presentation of analysis methods.

Results

Solid statistical analysis methods and presentation of results.

There are a lot of table but I’m not sure how to suggest minimizing them as they all contain pertinent info to the study.

Overall references look good, but I did not take a close look at each of them.

Figures and Tables look great!

6. PLOS authors have the option to publish the peer review history of their article (what does this mean?). If published, this will include your full peer review and any attached files.

Reviewer #1: Yes: JONATHON J DONAGER

Reviewer #2: No

---

## [Author Response · Author response to Decision Letter 0]

15 Oct 2019

Response to Reviewers for PONE-D-19180

Reviewer #1: General Comments

I enjoyed reading the manuscript and thought it well done. The comparison of models is interesting and the results are interesting and unique. I think the specific comparison of model input variables has the potential to reinforce the findings and provide some deeper ecological insight (see below). For example, which variables show up in both top ranking models? Why do you think that is in the system you examined? 

Thank you for your comments. See below for specific responses. Line numbers cited are from the Track Changes Version

I think placing figures 3 and 5 together (one below the other) would be useful to really compare the differences…

Done, now labelled as 3a and b.

Figure 4 needs larger text and more contrast, perhaps with different line patterns. (line type in R). It is hard to read.

Done (added colour and enlarged text).

Hypotheses need to be included in the final paragraph of your introduction. You make reference to them in the discussion.

We do not use the term hypothesis or hypotheses in the paper. The paper is not structured as a test of any hypotheses. It is about quantifying capacity of LiDAR to estimate understory. We re-read the discussion and don’t see any reference to hypotheses, although there is some discussion of the possible effect of occlusion. But this is more about anticipating what we might need to account for to generate good predictions rather than any kind of ecological hypothesis. We would prefer not to structure the paper in terms of hypotheses since the primary objectives do not fit well with that structure ie can we predict understory with LiDAR, are there some measures of understory that work better than others, and which modelling approach gives the best predictions. We did not have any a priori expectations on these three objectives. 

I would like to see some commentary on why you think the variables that floated to the top in both models were what they were. What do you think their ecological significance is? Or perhaps, why that understory structure shows up that way in the data. Do you think these models or indices would be relevant in other forests or ecosystems?

We included sentences in each section to indicate the significance of the variables, ie why they are likely important. It’s not generally an ecological reason. For example, it looks like STRATUM is important but not because of occlusion and maybe because of the improved ability to field sample vegetation in the lower strata. We modified the text to 

“These differences in prediction error suggest that the model can better predict new observations in the low stratum than the high stratum. A potential explanation for this result would be that the understory vegetation in the lower stratum is easier to estimate on the ground and therefore there is less noise in the relationship between the field and the LiDAR measures in the lower stratum.” Lines 387-391

Or for the understory LiDAR metrics…

We also modified the text to 

“All four LiDAR metrics were effective at predicting the amount of structure in an understory stratum, probably because they are all highly correlated direct measures of the density of understory vegetation. The best metric based on mixed effects modelling, however, was the voxel-based cover estimate (VOX1m) with the addition of STRATUM with a conditional R2 of 0.87. The voxel-based approach is relatively easy to calculate and provides a direct measure of the amount of understory structure. “ Lines 362-368

We added the following sentence to the discussion on RF models to address the additional variables included in the model:

“Other variables with high importance in the Random Forest models included other direct measures of understory structure, and canopy closure (S2 Table), which is expected to influence the amount of vegetation in the understory through light availability. The prediction error was slightly lower for the random forest model than for the mixed effects model (12.9% vs 15.6%), albeit at the cost of including 59 explanatory variables compared to 8 parameters estimated in the mixed effects model. Based on our results, generating landscape-wide predictions using the mixed-effects model should be simpler and more efficient than with the Random Forest model. For these reasons (12% higher explained variance, fewer explanatory variables, and similar prediction error), we recommend the mixed effects model for predicting understory vegetation structure with LiDAR. ” Lines 424-435

To address the generalizability of the models we added the following to the Conclusion:

“With increasing sampling density associated with better LiDAR technology, we anticipate that understory cover models will become more reliable and generalizable across regions. In particular, because the models are not dependent on any ecological relationships per se, because they use direct measures of vegetation cover, we believe that under similar sampling densities the models should be generalizable. Additional testing of this approach in different forested ecosystems would provide more confidence in the transferability of the models. “ Lines 498-504

Specific Comments

24 “… among other things” sounds very weak. I suggest changing this sentence to get a reader’s attention. Suggestion: “Forest understory vegetation is an important characteristic of the forest, but hard to measure with current remote sensing tools.”, or something along those lines.

We modified the first two sentences of the abstract to reflect this comment: see below for corrected sentences. 

“Forest understory vegetation is an important characteristic of the forest. Predicting and mapping understory is a critical need for forest management and conservation planning, but it has proved difficult with available methods to date.” Lines 24-27

40 If the random forest model had lower error, why did you choose the mixed effects model? Hope to return to this later.

The error in the random forest was not much lower than the one for the mixed effects model, whereas the variance explained by the mixed-effects model was much higher (12%) and used 51 fewer explanatory variables. For these reasons, we choose the mixed effects model, but we also clearly stated that the RF model was good too. We modified the text to clarify this point. 

“Our selected Random Forest model had 59 explanatory variables, whereas the best mixed effects model had two explanatory variables and their interaction, as well as a random effect of plot. Other variables with high importance in the Random Forest models included other direct measures of understory structure, and canopy closure (S2 Table), which is expected to influence the amount of vegetation in the understory through light availability. The prediction error was slightly lower for the random forest model than for the mixed effects model (12.9% vs 15.6%), albeit at the cost of including 59 explanatory variables compared to 8 parameters estimated in the mixed effects model. Based on our results, generating landscape-wide predictions using the mixed-effects model should be simpler and more efficient than with the Random Forest model. For these reasons (12% higher explained variance, fewer explanatory variables, and similar prediction error), we recommend the mixed effects model for predicting understory vegetation structure with LiDAR. ” Lines 421-435

52 remove the word “potentially”, it is redundant.

Done as suggested

53 I would qualify this statement… Lidar itself is just data, the estimates come when models are created, which is what you are doing. Therefore, Lidar “can provide estimates” or something like that would be more apt. As you know, understory vegetation is typically obscured by the dominant canopy, which is why your models would be useful.

We reworded 

“Active remote-sensing technology such as LiDAR (light detection and ranging) could be used to generate estimates to address this issue.” Lines 54-56

56-58 The most important measurement here is the time of flight of the reflected pulse, not just the pulse itself, which would give you the intensity of the return, so that piece of information in the third sentence here should come earlier. I think these sentences could be combined into a more concise description.

We reworded

“LiDAR provides an estimate of the three-dimensional forest structure including estimates of canopy structure, understory vegetation, and terrain. LiDAR is a survey method that measures the return time of a laser light pulse reflecting off solid objects such as the vegetation or the ground. These laser returns generate a three-dimensional representation of the forest.”Lines 57-60

68-72 I don’t think you need to state these things here, as you repeat them in the remaining part of the introduction. If anything, these things should come at the very end of the introduction when you are tying everything together that you have presented thus far.

Okay, we have removed these statements as suggested.

77 Heterogeneity is also caused by topography and the vegetation structure itself. And you should definitely have some citations here. Here’s one off the top of my head:

Goodwin, N. R., Coops, N. C., & Culvenor, D. S. (2006). Assessment of forest structure with airborne LiDAR and the effects of platform altitude. Remote Sensing of Environment, 103(2), 140-152.

We are struggling with this comment a bit. We read the suggested paper but did not find the specific reference to topography and veg structure influencing sampling density. We found another reference that supported the topography idea but not the vegetation structure. We added the idea of topography influencing sampling density with the new citation. We are not sure that we understand how vegetation structure would influence sampling density. Vegetation structure influences point density, but what we were interested in here were the variables that create noise in the relationship between vegetation density and point density. 

82 I would include other citations here… there have been numerous efforts to normalize lidar point density. For example:

Ruiz, L., Hermosilla, T., Mauro, F., & Godino, M. (2014). Analysis of the influence of plot size and LiDAR density on forest structure attribute estimates. Forests, 5(5), 936-951.

And take a look at this one:

Jakubowski, M. K., Guo, Q., & Kelly, M. (2013). Tradeoffs between lidar pulse density and forest measurement accuracy. Remote Sensing of Environment, 130, 245-253.

We added these citations to the discussion on normalizing lidar point density

92 You need a citation here concerning machine learning.

We added Cutler et al. 2007 on using Random Forest in Ecology

93 Also a citation here.

We now cite Latifi et al. 2017 and Penner et al. 2013 (refs 23 and 24)

96 Citation here

We now cite De’ath 2000 on machine learning 

99 And citation here. To make declarative statements about model parameters and interpretability like these, you should be referencing something.

We now cite De’ath 2000 

118 What hypotheses do you have concerning variable importance, model fit, etc. that you could reference in the discussion?

We haven’t structured the paper in terms of hypothesis testing. The main objective is to evaluate the capacity of LiDAR to generate good predictions of understory vegetation cover. We could add some hypotheses in the sense of ideas about how best to generate those predictions but we see it as somewhat artificial to call these the hypotheses of the paper. So for example we don’t have a hypothesis about whether machine learning or mixed effects models will be better for prediction. We don’t have any hypotheses about which of the 4 direct measures of understory are likely to be best for prediction. We would prefer to structure the paper in terms of well stated objectives rather than hypotheses. 

144 I’d like to see something here concerning the justification for stratifying your plot locations based on the data that you are trying to predict with… it seems somewhat circular. I think it is ok, as that is really the only way you could come up with a stratification based on understory across a landscape, but you should still address this issue.

Our intent with the stratification was to fill the model prediction space as much as possible. We knew that sites with more overstory were on average likely to have less understory because of the limited light availability. But we were interested in whether or not occlusion would play a role in prediction accuracy so we needed to try to have sites with both lots of overstory and lots of understory or vice versa to fill in the modelling space. So rather than just using a random approach that would have skewed the sample to the common conditions, we were selective in trying to represent the more uncommon conditions. We used the only data we had to improve the sampling to capture the uncommon conditions. We don’t believe this created any bias in the data but it may not have been very effective if the LiDAR data was not at all representative of the actual understory vegetation condition. We have added a few sentences to the methods to acknowledge this reasoning. 

“We acknowledge that this stratification would not be effective if the relative number of LiDAR pulse returns was unrelatedto actual understory vegetation cover. However, it was the most intuitive method to ensure that all overstory and understory conditions were represented in the sample.” Lines 159-163

154 What was the mean/variation of the horizontal precision of your GPSed plot coordinates? Sub meter could mean 0.01m or 0.99m

We don’t have these data. The reference manual indicates sub meter accuracy but we did not test this. We think the important point is that error in GPS location will introduce error but even with this error we are able to make good predictions. We added a citation of the reference manual and added some text on the implications of GPS error to the discussion.

“In addition, the error associated with GPS locations can introduce error into the relationship between ground-based and LiDAR estimates, although GPS technology is constantly improving. Our GPS (SXblue), reports sub meter accuracy under ideal conditions, but probably accounts for some of the error in prediction.” Lines466-470

163 The diagram is very helpful in understanding the plot design.

Thank you.

180 Could you justify the temporal discrepancy here? I am wondering how much the understory might have changed in the intervening 3 and 4 years between lidar acquisition and data collection.

We can’t justify it. Ideally the ground plots should be sampled immediately after the LiDAR acquisition, but that was not possible for a variety of reasons. We have no way to quantify the change in the understory over the 3-4 years. Likely this is not very significant in mature forests, but might be in disturbed stands. In our case most stands were mature. We have discussed this as a likely source of error but one that, based on the results, did not undermine, to any great degree, the predictive capacity of LiDAR. 

189 I like this section. Easy to follow.

Thank you.

256 “appeared” is not a precise word. Perhaps something like “were slightly more linearly correlated”.

The difference in correlation values is very small and I would assume not significant in a statistical sense.

Okay we changed this

272 “For each of the four…”

Changes as suggested

275 This is an interesting finding!

296 This should be referencing back to your initial hypotheses, but you didn’t present any in your final introductory paragraph.

We reworded this sentence to remove the term expectations, and stated explicitly the implications from the result. We would prefer not to structure the paper in terms of hypothesis testing.

“The SMAPE value was lowest for the lowest strata (0.107) and greatest for the highest strata (0.190) suggesting no evidence of occlusion.”Lines 314-316

299 The labels on this figure should be larger.

We enlarged the text

321 based on the lower error rate and looking at the scatter in Figure 5, I would say that the random forest model did a much better job in predicting understory strata.

Both models did well. We selected the mixed effects model because of the much higher (12%) variance explained. The difference in prediction error was relatively low (<3%). Also random forest used 59 predictors vs the 2 plus interaction for the mixed effects model. We modified the text to clearly justify our reasoning but also added an acknowledgement that the RF model was also effective.

“Based on our results, generating landscape-wide predictions using the mixed-effects model should be simpler and more efficient than with the Random Forest model. For these reasons (12% higher explained variance, fewer explanatory variables, and similar prediction error), we recommend the mixed effects model for predicting understory vegetation structure with LiDAR, but we acknowledge that the Random Forest model also generates robust predictions.” Lines 429-435

331 I would like to see the ranking of the most important variables in your final random forest model.

We have added them to the supplementary material and added an S Table citation in the results to that effect. 

345 This is from your mixed effects model. I personally think the random forest model did a better job of prediction, although as you state in the introduction, not as easy to interpret as the linear model. But when using such rich data and derivatives as possible with lidar, you might as well have a model that includes all relevant information, such as a random forest model (in my opinion, you might disagree). Perhaps here though, you could compare which variables were included in your top ranking linear models to the most important ranked variables in your final random forest model. Is there overlap? I think this would not only serve to justify the models and variables, but to compare what the models themselves say about the relevant predictor variables.

We have added the variable importance from the selected RF model and made reference to it in the Results and Discussion. We highlight the overlap in variable selection between the two models in the Discussion paragraph on the Random Forest model. 

348 Your hypothesis should come before the discussion and reference them here.

We have not structured the paper around specific hypotheses because the main objective it to test the capacity of LiDAR to predict understory vegetation which would only generate a trivial hypothesis that LiDAR can or can’t do the job. The idea that occlusion might be important could be worded as a hypothesis but we have evidence from the literature to suggest both options so again the hypothesis would be somewhat artificial. We are trying to take the focus away from somewhat arbitrary hypotheses and predictions and focus the paper as an evaluation of LiDAR as a tool. 

389 You don’t mention random forest models up until this point. All of your discussion thus far comes off as there only being one model type explored… I would mention random forest earlier and like you did for mixed effects, discuss the final model and the most important variables within that model.

We are going to resist this suggestion as it would require a very significant reworking of the discussion that we do not feel is warranted. Our primary objective is to quantify the capacity of LiDAR to estimate understory structure so that is what we focus on in the first paragraphs of the discussion. The comparison of modelling approaches is identified in the introduction as our third objective and so we maintained that structure in the discussion. If we had found that the RF model was significantly better at estimating understory we might have reorganized the introduction but because we don’t see a big advantage of the RF model over the mixed-effects model, our discussion of the mixed effects models results fully answers our first objective. It is also a better model for examining objective 2 and so we believe the discussion flows as it should and in parallel to the intro/objectives.

396 Variance explained doesn’t matter as much as the error… the random forest model had a lower rate of error and that should be stated here to counterbalance the statement about variance.

We believe that the variance explained is a good measure of the capacity of the model to explain/predict the ground based measures of understory cover. As well the variance explained was 12% higher in the mixed effects model but the error was 2.7 percent lower in the RF model. We see these error estimates as being very similar. But there are other reasons including the simplicity of only needing 2 variables in the mixed effects model vs 59 in the RF model that led us to recommend the ME model. We added additional justification and an acknowledgement of the value of the RF model. We think we are justified in recommending the ME model but the reader is free to use the RF model and we acknowledge that it will produce good predictions. 

“The comparisons of mixed effects and Random Forest models revealed some obvious alignment. All four of the LiDAR metrics considered (fractional cover, leaf area density, normalized cover, and voxel cover) produced models with high R2 values. All four of these variables also had very high variable importance in the Random Forest models. Voxel cover (VOX1m) was the most important variable in the selected Random Forest Model. The stratum variable appeared often in the top Random Forest models and was also important in the top-ranked mixed-effects model (VOX1m * STRATUM). The Random Forest model had a high variance explained (75%), but not as high as the best mixed effects model that included the voxel-based measure of cover (87%). Our selected Random Forest model had 59 explanatory variables, whereas the best mixed effects model had two explanatory variables and their interaction, as well as a random effect of plot. Other variables with high importance in the Random Forest models included other direct measures of understory structure, and canopy closure (S2 Table), which is expected to influence the amount of vegetation in the understory through light availability. The prediction error was slightly lower for the random forest model than for the mixed effects model (12.9% vs 15.6%), albeit at the cost of including 59 explanatory variables compared to 8 parameters estimated in the mixed effects model. Based on our results, generating landscape-wide predictions using the mixed-effects model should be simpler and more efficient than with the Random Forest model. For these reasons (12% higher explained variance, fewer explanatory variables, and similar prediction error), we recommend the mixed effects model for predicting understory vegetation structure with LiDAR, but we acknowledge that the Random Forest model also generates robust predictions.” Lines 413-435

428-431 Yes! This is what I was thinking when I read the methods. I think you need to expand on this. Would your results been similar if they were collected during the same year? I think this is a really important point and shouldn’t just come at the end of a paragraph.

We have added more content to the paragraph on error. We acknowledge that there is some variation in the predictability and identify 2 sources of noise in our data that may affect other studies.

“It is unclear why there is so much variation in the ability of LiDAR to predict understory structure but it suggests that we should be somewhat cautious in assuming that individual LiDAR metrics are always capturing the understory structure. It is important to note that some of the error in prediction in our models is likely the result of the lag between the LiDAR acquisition (2012) and the field data acquisition (2016-2017). This lag is likely to result in the most error in the youngest stands where changes in herb and shrub growth are likely to be greatest but I in the analysis, most stands are mature forest. Likely with less lag between LiDAR and ground-based measures we would have seen even better predictions. In addition, the error associated with GPS locations can introduce error into the relationship between ground-based and LiDAR estimates, although GPS technology is constantly improving. Our GPS (SXblue), reports sub meter accuracy under ideal conditions, but discrepancy in geoposition probably accounts for some of the error in prediction.” Lines 458-470

443 You talk a lot about the data and the models… I would like to see some hypothesis building about why certain metrics were better and how they are directly related to the structure.

We are not comfortable with a lot of hypothesis building for this paper as we had few a priori expectaitons about the outcome. We have provided justification throughout these responses to argue that, in the end, the models don’t provide a great deal of ecological insight but do effectively demonstrate the value of LiDAR to capture understory vegetation structure. 

443 I would also like to know, as a scientist, how you think these models, indices, metrics, etc. might work in other forests? Do you think they are particular to this ecosystem?

We added the following text to the conclusions to suggest that we think the models are transferable but it would be good to test them in some new forested ecosystems.

“With increasing sampling density associated with better LiDAR technology, we anticipate that understory cover models will become more reliable and generalizable across regions. In particular, because the models are not dependent on any ecological relationships per se, because they use direct measures of vegetation cover, we believe that under similar sampling densities the models should be generalizable. Additional testing of this approach in different forested ecosystems would provide more confidence in the transferability of the models. “ Lines 498-504

Reviewer #2

Abstract

Line 27: has yet to be fully validated. 

Changed as requested.

Introduction

Line 86 (and as it occurs thereafter): some papers cite Random Forests with capital letters. I think it is fine either way.

We converted all instances of “random forest” to “Random Forest”.

Excellent lit review. The authors did an especially good job exploring various options for analysis.

Thank you.

Methods

Line 126: extra space between composition features

Corrected.

Thorough Methods section. I like Figure 1.

Thank you.

Analysis

Line 214: insert space between variables (26)

Added as requested.

Line 229: insert space between package (18…

Added as requested.

Thorough presentation of analysis methods.

Thank you.

Results

Solid statistical analysis methods and presentation of results.

There are a lot of table but I’m not sure how to suggest minimizing them as they all contain pertinent info to the study.

We preferred to maintain the tables in their original form to convey the important information.

---

## [Decision Letter · Decision Letter 1]

7 Nov 2019

Modelling vegetation understory cover using LiDAR metrics

PONE-D-19-19180R1

Dear Dr. Venier,

We are pleased to inform you that your manuscript has been judged scientifically suitable for publication and will be formally accepted for publication once it complies with all outstanding technical requirements.

With kind regards and congratulations,

John Toland Van Stan II, Ph.D.

Academic Editor

PLOS ONE

Reviewers' comments:

Reviewer's Responses to Questions

**Comments to the Author**

1. If the authors have adequately addressed your comments raised in a previous round of review and you feel that this manuscript is now acceptable for publication, you may indicate that here to bypass the “Comments to the Author” section, enter your conflict of interest statement in the “Confidential to Editor” section, and submit your "Accept" recommendation.

Reviewer #1: All comments have been addressed

Reviewer #2: All comments have been addressed

2. Is the manuscript technically sound, and do the data support the conclusions?

Reviewer #1: Yes

Reviewer #2: Yes

3. Has the statistical analysis been performed appropriately and rigorously? 

Reviewer #1: Yes

Reviewer #2: Yes

4. Have the authors made all data underlying the findings in their manuscript fully available?

Reviewer #1: Yes

Reviewer #2: Yes

5. Is the manuscript presented in an intelligible fashion and written in standard English?

Reviewer #1: Yes

Reviewer #2: Yes

6. Review Comments to the Author

Reviewer #1: I am very pleased with the responses to my previous comments and think the manuscript is of high quality. The manuscript satisfies all of the above criteria. I support publication of this manuscript.

Reviewer #2: I am please with the authors' revisions and response. I have no further comments. I would move to accept the manuscript.

7. PLOS authors have the option to publish the peer review history of their article (what does this mean?). If published, this will include your full peer review and any attached files.

Reviewer #1: Yes: Jonathon J Donager

Reviewer #2: No

---

## [Editor Report · Acceptance letter]

12 Nov 2019

PONE-D-19-19180R1 

Modelling vegetation understory cover using LiDAR metrics 

Dear Dr. Venier:

I am pleased to inform you that your manuscript has been deemed suitable for publication in PLOS ONE. Congratulations! Your manuscript is now with our production department. 

With kind regards,

on behalf of

Dr. John Toland Van Stan II 

Academic Editor

PLOS ONE